# High Yield Synthesis of Curcumin and Symmetric Curcuminoids: A “Click” and “Unclick” Chemistry Approach

**DOI:** 10.3390/molecules28010289

**Published:** 2022-12-30

**Authors:** Marco A. Obregón-Mendoza, William Meza-Morales, Yair Alvarez-Ricardo, M. Mirian Estévez-Carmona, Raúl G. Enríquez

**Affiliations:** 1Instituto de Química, Universidad Nacional Autónoma de México, Circuito Exterior, Ciudad Universitaria, Ciudad de México 04510, Mexico; 2Departamento de Farmacia, Escuela Nacional de Ciencias Biológicas, Instituto Politécnico Nacional, Wilfrido Massieu SN, U. A. Zacatenco, Ciudad de México 07738, Mexico

**Keywords:** curcumin, *Curcuma longa*, bis-demethoxycurcumin, Michael acceptor, diferuloylmethane, BF_3_·THF, alumina

## Abstract

The worldwide known and employed spice of Asian origin, turmeric, receives significant attention due to its numerous purported medicinal properties. Herein, we report an optimized synthesis of curcumin and symmetric curcuminoids of aromatic (bisdemethoxycurcumin) and heterocyclic type, with yields going from good to excellent using the cyclic difluoro-boronate derivative of acetylacetone prepared by reaction of 2,4-pentanedione with boron trifluoride in THF (ca. 95%). The subsequent cleavage of the BF_2_ group is of significant importance for achieving a high overall yield in this two-step procedure. Such cleavage occurs by treatment with hydrated alumina (Al_2_O_3_) or silica (SiO_2_) oxides, thus allowing the target heptanoids obtained in high yields as an amorphous powder to be filtered off directly from the reaction media. Furthermore, crystallization instead of chromatographic procedures provides a straightforward purification step. The ease and efficiency with which the present methodology can be applied to synthesizing the title compounds earns the terms “click” and “unclick” applied to describe particularly straightforward, efficient reactions. Furthermore, the methodology offers a simple, versatile, fast, and economical synthetic alternative for the obtention of curcumin (85% yield), bis-demethoxycurcumin (78% yield), and the symmetrical heterocyclic curcuminoids (80–92% yield), in pure form and excellent yields.

## 1. Introduction

Curcumin [(1,7-bis-(4-hydroxy-3-methoxy-phenyl)-1,6-heptadien-3,5-dione] [1,2], also known as diferuloylmethane, is a bioactive molecule found in the rhizome of the *Curcuma longa* Asian spice from *Zingiberaceae* family [3,4,5,6,7,8,9,10,11,12,13,14,15,16,17,18,19,20] Chemically, curcumin is described as a typical Michael-type acceptor [21,22,23,24]. The worldwide scientific interest in this molecule is due to its broad therapeutic activities, which include its purported properties as an anticancer [25,26,27,28,29,30,31], antiangiogenic disease [32], antimetastatic [33], antioxidant [34,35,36], free radical scavenger [37], anti-inflammatory [38,39], antidepressant [40] and anti-Alzheimer’s disease agent [41,42,43,44,45].

The importance of having curcumin as a pure metabolite lies in expanding its complementary studies on its pharmacokinetics, pharmacodynamics, and toxicological studies [46], which help to understand the effects of this fascinating bioactive molecule in living systems. The research in chemical synthesis has been mainly directed to the preparation of derivatives [47,48] that help overcome its physicochemical properties [49] and rapid metabolism [27,42,50] and to increase its bioavailability [11,51].

Curcumin has a long historical pathway as a natural colorant [49,52,53,54] (E100), and curcumin’s chemistry began early in the 19th century. Curcumin was isolated in 1815 [9], and its crystallized form was known in 1870 [54]. Until early in the 20th century, the molecular formula of curcumin (C_21_H_20_O_6_) could be assessed by Miłobȩdzka, J. et al. in 1910 [55]. In the following years, Lampe [56] (1918), Pavolini [54] (1950), and Pabon [57] (1964) carried out a total synthesis of this molecule. The chemical structure was confirmed in 1973 [58], while studies of the keto-enol equilibrium [3,41,59,60,61], stability [62], and metabolic pathway [63,64,65] in vivo stand out as relatively recent findings.

Although separations of curcumin are reported in various investigations [8,10,13,66,67], its purification is a difficult task due to the presence of two closely related curcuminoids (i.e., demethoxycurcumin and bis-demethoxycurcumin [25,26,53,68,69]) due to co-crystallization phenomena. Furthermore, obtaining high-purity curcumin from natural sources is difficult since it involves repeated chromatographic and crystallization procedures.

In the methodologies developed for synthesizing curcumin, it was early recognized that protecting the α-diketone functionality is a critical step for the subsequent condensation of two vanillin molecules at the sidechain methyl groups. Thus, the well-known secondary Knoevenagel reaction on carbon C-1 is avoided [54]. In addition, adequate protection of the b-diketone function can be achieved through boron complexes using reagents such as boron trioxide [47,57,70], boric acid [58], and, more recently, boron trifluoride [71]. The approach named “click chemistry” is applied in the obtention of compounds following simple steps of joining small modular units [72]. In the present case, the protective reaction on the b-diketo function to give the BF_2_ derivative occurs with a high degree of efficiency, i.e., in a “click” fashion. Furthermore, the removal of the BF_2_ group occurs under equally simple conditions and high efficiency, allowing us to propose the term “unclick” for this reaction step.

The synthetic approach used in our work is adequate for obtaining the natural symmetric curcuminoids curcumin and bisdemethoxycurcumin, with a significant reduction of expensive chromatographic and crystallization steps. However, the other essential natural asymmetric demethoxycurcumin requires a somewhat different synthetic route, which is under investigation. Nevertheless, the method demonstrated robustness for synthesizing symmetric heterocyclic curcuminoids using the corresponding aldehydes. A convenient feature altogether is the economy of reagents and laboratory steps needed.

## 2. Results

Although the protection reaction of acetylacetone is commonly carried out with boron trifluoride etherate [73,74,75], its manipulation requires extreme caution [2]. A much safer alternative is found using boron trifluoride complex in THF (Figure 1). Five advantages at least are introduced, i.e., (I) minimum release of toxic vapors from the container, (II) both high density (1.268 g/mL) and boiling point (180 °C) allows easier manipulation when measuring the required volumes; (III) the addition of the reagent to the reaction flask is carried out at room temperature; (IV): no violent reaction is observed upon addition of reagents and (V): the use of inert atmosphere does not seem critical for the reaction to proceed.

The synthesis of curcuminoids-BF_2_ has been previously reported in a one-pot reaction [76]. In our scaled-up approach (98 mmol), it was found convenient a stepwise procedure to overcome the bulk generation of HF, which promotes the formation of quaternary ammonium salts from *n*-butylamine. The isolation of a powdered product renders a rather convenient material for further workup. Thus, the BF_2_ derivatives can be advantageously manipulated and purified as solid starting materials, favoring cleaner and higher overall yields (see Table 1).

The high yields obtained in the aldol condensation reaction (Figure 2) are explained by the following two reasons: (1) the precipitation of the condensed compound consequently produces a continuous consumption of the reactants in solution [76] and (2) the protection of acac through the use of BF_3_ is an approach that affords much better yields [77].

The crude product of the aldol condensation reaction to obtain curcumin-BF_2_ contains residues (see Figure 1) of *n*-butylamine and tributyl borate, which are easily removed after washing with a mixture of distilled water and acetone (10–20% acetone), see Figure 2.

One of the critical steps in the synthesis of curcuminoids (heptanoids) is the cleavage of the BF_2_ group to obtain 1,3-diketone form (or enol). Yields greater than 80% are reported when the BF_2_ group is hydrolyzed in several media (organics: MeOH/DMSO [76,78] and MeOH/DMSO/triethylamine or inorganics: diluted NaOH [73] and sodium oxalate [79]). However, the efficiency and reproducibility of reported procedures have been considered limited [73].

The removal of boron reaction by-products has been reported using inorganic salts [80] (e.g., aluminum sulfates and sodium aluminates) or silica, but efficient removal has been reported using amorphous Al_2_O_3_ [81]. This feedback has served to assay additional means that can catalyze the hydrolysis of the BF_2_ group through the use of three different metal-hydrated oxides (Figure 3): SiO_2_ (silica) or Na_12_[(AlO_2_)_12_(SiO_2_)_12_]_·_xH_2_O (molecular sieves) or Al_2_O_3_ xH_2_O (alumina). 

Initially, it was chosen to carry out the opening reactions catalyzed in silica using two different alcoholic solvents (ethanol and methanol). However, ethanol is more eco-friendly, and curcumin was obtained 72 h later in low yield (possibly due to the adsorption of curcumin to silica). Therefore, methanol was found more appropriate for removing the boron-difluoride moiety (Table 2).

## 3. Discussion

The synthesis of curcumin and curcuminoids has been carried out with three simple reaction steps: (1) protection of keto-enol functionality of acetylacetone (acac) by BF_3_·THF; (2) condensation of the corresponding aromatic aldehyde catalyzing with *n*-butylamine; (3) cleavage of the BF_2_ group by means of hydrated metal oxides. Curcumin, bis-demethoxycurcumin itself, and two heterocyclic curcuminoids were obtained with very good yields and were fully characterized by spectroscopic techniques.

In a general description, this procedure consists of three basic yet simple general steps: (a) a protective step (reaction of the 2,4-pentanedione with boron trifluoride avoiding the Knoevenagel secondary reaction) while activating the methyl groups promoting (b) the efficient aldol condensation and (c) the deprotecting reaction step removing the BF_2_ group and allowing the recovery of the original b-diketone function.

It suggested that the mechanism for the removal of the BF_2_ group is due to an anion exchange phenomenon involving the reaction of boron and the basic OH—group or water in agreement with previous mechanistic proposals [81,82], which are specifically adsorbed and are present at the surfaces of hydrated metal oxides [83,84,85]. A possible reaction mechanism is depicted in Figure 4.

The proposal mechanism in step I is supported by Venkata [71], and steps II, III, and IV are supported by Weiss [73]. Adsorption and removal of the B(OH^−^)_4_ species are supported by previous references [80,81,82,83,84].

The integrity of the free curcuminoid on the aluminum oxides or under the reaction medium does not lead to decomposition since it is known that the breakdown of the BF_2_ generates HF [73], and the curcuminoids are relatively unaffected by pH from 2 to 7 [8]. Furthermore, greater boron adsorption by alumina occurs between pH 6 and 7 [81,86].

Our best yields in the obtention of curcumin were achieved using MeOH/Al_2_O_3_, probably associated with the more significant boron adsorption in acidic pH, though other authors associate boron adsorption with the presence of hydroxide ions present in alumina [86].

The ^1^H-NMR spectrum of curcumin (Figure 3) confirms the assigned structure, and characteristic signals of the vinyl protons (α,β-unsaturated, system AB) are present in the form of two doublets at 7.54 and 6.75 with coupling constants ca. 16 Hz (trans). Evidence for the keto-enol tautomerism is given by the signal observed at 16.47 ppm (enol) and the signal corresponding to the methine proton (CH) at 6.06 ppm. Additionally, the DEPT-135 spectrum (see Appendix A) shows no (CH_2_) methylene carbons; methines (CH) and methyl groups (CH_3_) are observed as positive signals and fit satisfactorily with data reported in the literature [59]. Similarly, the ^1^H-NMR spectra of all other symmetric curcuminoids show a consistent correlation between structure and spectral features (see Appendix A).

The mass spectrum (MS) of curcumin shows a characteristic peak at *m*/*z* = 368, which corresponds adequately to the molecular ion of curcumin and is consistent with the chemical formula C_21_H_20_O_6_. In addition, the spectrum shows a base peak with *m*/*z* = 177 representing the expected molecular fragment. Mass spectra of bis-demethoxycurcumin (6) *m*/*z* = 308, furan-curcumin (7) *m*/*z* = 256, and thiophene-curcumin (8) *m*/*z* = 288 show a consistent peak with the chemical formulas C_19_H_16_O_4_, C_15_H_12_O_4_, and C_15_H_12_O_2_S_2_, respectively (see Appendix A).

The present synthetic route was successfully extended for the obtention of other symmetrical curcuminoids (compounds **6**–**8**) with 4-hidroxybenzaldehyde, furfural, and thiophenecarboxaldehyde. Thus, when 4-hidroxybenzaldehyde and furfural were used in the corresponding curcuminoid synthesis using a modified Pabon´s approach, the yields decreased significantly to a reported 33 and 8%, respectively [77,87].

Interestingly, the heterocyclic curcuminoid resulting from 2-thiophene carboxaldehyde (compound **4**) afforded excellent yields (95%) in the aldol condensation reaction, while the cleavage of the BF_2_ group on MeOH/alumina afforded (compound **8)** in 92% yield. This overall high yield is even higher than the corresponding one observed for curcumin.

The term “Click Chemistry” [88] has been adopted for curcumin synthesis based on three simple concepts: (I) reactions are broad in scope and give high yields; (II) starting reagents are readily available, and simple reaction conditions are needed, and (III) no chromatographic methods are required to purify curcumin and other symmetric curcuminoids. The term “Unclick” refers to the efficient removal of the protecting/activator group, namely BF_2_, which was also achieved in high yield.

## 4. Materials and Methods

Boron trifluoride.THF complex (CAS 462-34-0), silica gel high-purity grade, average pore size 60 Å (52–73 Å), 70–230 mesh, 63–200 μm, for column chromatography (CAS 112926-00-8), molecular sieves 4Å beads, 8–12 mesh (CAS 70955-01-0) and Alumina Brockmann III (1344-28-1) were purchased from Sigma-Aldrich and were used without prior activation or purification.

All chemicals were available commercially, and the solvents were purified with conventional methods before use [89].

Melting points were determined on an Electrothermal Engineering IA9100 digital melting point apparatus in open capillary tubes and were uncorrected [1,2].

^1^H and ^13^C NMR spectra were obtained in a Bruker Fourier 400 MHz spectrometer using TMS as an internal reference and CDCl_3_ or Acetone-*d_6_*, or DMSO-*d_6_* as solvents. NMR spectra were processed with MestreNova software 12.0.0 [90] and are found in the Appendix A.

Spectroscopic measurements. IR absorption spectra were recorded using an FT-IR Bruker Tensor 27 spectrophotometer in the range of 4000–400 cm^−1^ as KBr pellets [1,2] (see Appendix A).

Mass Spectra were recorded using The MStation JMS-700 JEOL equipment (Electron Ionization, 70 eV, 250 °C, Impact positive mode and calibration standard with perfluorokerosene) and the AccuTOF JMS-T100LC JEOL equipment (DART^+^, 350 °C, positive ion mode and calibration standard with PEG 600) [1,2]. All mass spectra are shown in Appendix A.

HPLC chromatograms were obtained using an Agilent 1260 infinity II with diode -UV detector at 425 nm, column Eclipse Plus C18(2) 100 × 2.0 mm 3 μm; eluting with a solvent gradient (previously described with minor modifications [70,91]) from acetonitrile/water (acetic acid 2%) 40:60 to acetonitrile/water (acetic acid 2%) 50:50 and are included in the Appendix A.

### 4.1. Synthon Preparation

In a 250 mL round flask, 10 mL of 2,4-pentanodione (acac, 98 mmol) was dissolved in 30 mL of dichloromethane; subsequently, 11 mL of boron trifluoride tetrahydrofuran complex (BF_3_·THF, 98 mmol) was added to the solution, and the reaction was left overnight with magnetic stirring at room temperature. After, the organic phase was concentrated in vacuo affording the resulting product, which can be directly used for the following reaction step.

2,2-difluoro-4,6-dimethyl-2H-1,3,2-dioxaborinin-1-ium-2-uide (*Synthon*): yield 95%, solid amber, melting point 40 °C, ^1^H NMR (400 MHz, CDCl_3_, TMS): δ 5.96 (s, 1H, Methine-H), 2.27 (s, 6H, Methyl-H); ^13^C NMR (100 MHz, CDCl_3_, TMS): δ 192.63 (C=O), 102.12 (C_methine_-H), 24.32 (-CH_3_). IR (KBr) 1556 v(C=O, C=C), 1148 v(B-F, B-O), 1086 v(B-F, B-O)cm^−1^, DART^+^-MS: *m/z* (%) = 129 (148–19), *m/z* calc. = 148.

### 4.2. Condensation of Aldehydes with the Synthon

The curcuminoid-BF_2_ symmetric structure is obtained by an aldol condensation reaction under similar experimental conditions previously reported [71,73,74,75,76,77].

### 4.3. General Methodology

Mixture 1. In a 100 mL Erlenmeyer flask, 7.5 g of vanillin (49 mmol) was dissolved in 25 mL of EtOAc, 6.3 mL of tributyl borate (24.5 mmol) was added, and this mixture was heated until homogenization was achieved.

Mixture 2. In a 250 mL round flask, 4 g of *synthon* (1.1 eq, 27 mmol) was dissolved in 25 mL of EtOAc, and then the homogenous product 1 was added to the solution; then, 2.7 mL of N-butylamine (27 mmol, in 10 mL of EtOAc) was added dropwise. The reaction was left overnight with magnetic stirring at room temperature. Finally, a solid red precipitate was filtered-off and washed with a mixture of 50 mL water/acetone 90::10. This same methodology (same molar amounts) was carried out to synthesize the symmetrical curcuminoids-BF_2_.

Spectral data of Curcumin-BF_2_ (1): 2,2-difluoro-4,6-bis((*E*)-4-hydroxy-3-methoxystyryl)-2*H*-1,3,2-dioxaborinin-1-ium-2-uide, yield 90%, red solid, melting point 230 °C, ^1^H NMR (400 MHz, DMSO-*d*_6_, TMS): δ 10.10 (s, 2H,-OH), 7.92 (d, *J* = 15.6 Hz, 2H, Vinyl-H), 7.47 (d, *J* = 1.8 Hz, 2H, Aryl-H), 7.34 (dd, *J* = 8.3 Hz, 2H, Aryl-H), 7.02 (d, *J* = 15.6 Hz, 2H, Vinyl-H), 6.88 (d, *J* = 8.2 Hz, 2H, Aryl-H), 6.45 (s, 1H, Methine-H), 3.85 (s, 6H, -OCH_3_); ^13^C NMR (100 MHz, DMSO-*d*_6_, TMS): δ 178.72 (C=O), 151.34 (C-OH), 148.17 (C_aryl_), 146.97 (C_vinyl_-H), 125.99 (C_aryl_), 125.26 (C_aryl_-H), 117.86 (C_vinyl_-H), 115.95 (C_aryl_-H), 112.39 (C_aryl_-H), 101.12 (C_methine_-H), 55.76 (-OCH_3_). IR (KBr) 3482 v(-OH), 1615 v(C=O), 1586 v(C=C), 1509 v(C=O, C=C), 1146 v(B-F, B-O) cm^−1^, DART^+^-MS: *m/z* = 397 (416–19), *m/z* calc. = 416.

4-hydroxy-curcuminoid-BF_2_ (2) 2,2-difluoro-4,6-bis((*E*)-4-hydroxystyryl)-2*H*-1,3,2-dioxaborinin-1-ium-2-uide, yield 85%, red powder, melting point 225 °C, ^1^H NMR (400 MHz, Acetone-*d*_6_, TMS): δ 9.29 (br, 2H, -OH), 7.96 (d, *J* = 15.6 Hz, 2H, Vinyl-H), 7.73 (m, 4H, Aryl-H), 6.97 (m, 4H, Aryl-H), 6.90 (d, *J* = 15.6 Hz, 2H, Vinyl-H), 6.39 (s, 1H, Methine-H ); ^13^C NMR (100 MHz, Acetone-*d*_6_, TMS): δ 180.81 (C=O), 162.16 (C-OH), 147.35 (C_vinyl_-H), 132.67 (C_aryl_-H), 127.27 (C_aryl_), 118.96 (C_vinyl_-H), 117.15 (C_aryl_-H), 102.20 (C_methine_-H). IR (KBr) 3422 v(-OH), 1598 v(C=O), 1579 v(C=C), 1518 v(C=O, C=C), 1147 v(B-F, B-O) cm^−1^, EI-MS: *m/z* = no observed, *m/z* calc. = 356.

Furan-curcuminoid-BF_2_ (3) 2,2-difluoro-4,6-bis((*E*)-2-(furan-2-yl)vinyl)-2*H*-1,3,2-dioxaborinin-1-ium-2-uide, yield 80%, red powder, melting point 200 °C, ^1^H NMR (400 MHz, CDCl_3_, TMS): δ 7.64 (d, *J* = 15.15, 2H, Vinyl-H), 7.51 (d, *J* = 1.74, 2H, Aryl-H), 6.75 (d, *J* = 3.49, 2H, Aryl-H), 6.51 (d, *J* = 15.22, 2H, Vinyl-H), 6.47 (dd, *J* = 3.51, 1.77, 2H, Aryl-H), 5.96 (s, 1H, Methine-H); ^13^C NMR (100 MHz, CDCl_3_, TMS): δ 178.98 (C=O), 151.01 (C_aryl_), 146.78 (C_aryl_-H), 131.91 (C_vinyl_-H), 119.04 (C_aryl_-H), 117.95 (C_vinyl_-H), 113.37 (C_aryl_-H),102.34 (C_methine_-H). IR (KBr) 1619 v(C=O), 1569 v(C=O, C=C), 1455 v(C-H), 1385 v(C-H), 1276 v(C-O), 1067 v(B-F, B-O) cm^−1^, DART^+^-MS: *m/z* = 305 (304 + 1), *m/z* calc. = 304.

Thiophene-curcuminoid-BF_2_ (4) 2,2-difluoro-4,6-bis((*E*)-2-(thiophen-2-yl)vinyl)-2*H*-1,3,2-dioxaborinin-1-ium-2-uide, yield 95%, violet powder, melting point 270 °C, ^1^H NMR (400 MHz, Acetone-*d*_6_, TMS): δ 8.20 (d, *J* = 15.38, 2H, Vinyl-H), 7.84 (d, *J* = 5.10, 2H, Aryl-H), 7.73 (d, *J* = 3.70, 2H, Aryl-H), 7.26 (dd, *J* = 5.05; 3.69, 2H, Aryl-H), 6.79 (d, *J* = 15.38, 2H, Vinyl-H), 6.51 (s, 1H, Methine-H); ^13^C NMR (100 MHz, Acetone-*d*_6_, TMS): δ 180.71 (C=O), 140.92 (C_aryl_), 139.92 (C_vinyl_-H), 135.63 (C_aryl_-H), 133.32 (C_aryl_-H), 130.14 (C_aryl_-H), 120.67 (C_vinyl_-H), 102.81(C_methine_-H). IR (KBr) 1594 v(C=O), 1541 v(C=O), 1494 (C=C), 1411(-C=C_ring_), 1291 v(C-O), 1152 v(B-F, B-O) cm^−1^, DART^+^-MS: *m/z* = 317 (336–19), *m/z* calc. = 336. 

### 4.4. Reaction Conditions for “Unclick” Removal of the BF_2_ Group

In a 500 mL round flask, 10 g of curcuminoid-BF_2_ was dissolved in 400 mL of methanol (MeOH), 20% weight of metal oxide (catalyst) was added to the solution, and the mixture was left overnight under magnetic stirring at reflux. The reaction was quenched by filtration using a sintered glass funnel packed with celite. MeOH was evaporated in vacuo, and reaction crude was extracted with 150 mL of EtOAc (ethyl acetate) and water (3 × 100 mL). The organic phase was dried with Na_2_SO_4_ and concentrated in vacuo to afford the curcuminoid product, which was purified by recrystallization using EtOAc and hexane. The yields obtained for the synthesis of curcumin with several catalyzers were as follows: silica (70%), molecular sieves (80%) and alumina (85%). This same methodology (same amounts) was carried out for the synthesis of symmetrical curcuminoids (compounds **6**–**8**).

Curcumin (5) 1,7-bis-(4-hydroxy-3-methoxy-phenyl)-1,6-heptadien-3,5-dione, yellow-orange powder, purified by recrystallization using EtOAc and hexane, purity by HPLC 99.48%, melting point 180 °C, ^1^H NMR (400 MHz, DMSO-*d*_6_, TMS): δ 16.47 (br, 1H, Enol-H), 9.66 (br, 2H, -OH), 7.55 (d, *J* = 15.8 Hz, 2H, Vinyl-H), 7.32 (d, *J* = 1.89 Hz, 2H, Aryl-H), 7.15 (dd, *J* = 8.2; 1.93 Hz, 2H, Aryl-H), 6.82 (d, *J* = 8.13 Hz, 2H, Aryl-H), 6.75 (d, *J* = 15.81 Hz, 2H, Vinyl-H), 6.06 (s, 1H, Methine-H), 3.84 (s, 6H, -OCH_3_); ^13^C NMR (100 MHz, DMSO-*d*_6_, TMS): δ 183.22 (C=O), 149.36 (C-OH), 148.00 (C_aryl_), 140.72 (C_vinyl_-H), 126.34 (C_aryl_), 123.14 (C_aryl_-H), 121.10 (C_vinyl_-H), 115.70 (C_aryl_-H), 111.33 (C_aryl_-H), 100.85 (C_methine_-H), 55.69 (-OCH_3_). IR (KBr) 3506 v(OH), 1628 v(C=O), 1602 v(C=C_ring_), 1509 v(C=O, C=C), 1428 v(C-O_phenol_), 1281 v(C-O_enol_), 1154 v(C-O), 1028 v(=C-O-CH_3_) cm^−1^, EI-MS: *m/z* = 368, *m/z* calc. = 368.

Bis-demethoxycurcumin (6) 1,7-bis(4-hydroxyphenyl)-1,6-heptadien-3,5-dione, red-orange powder, purified by recrystallization using CH_2_Cl_2_ and MeOH, purity by HPLC 99.45%, melting point 215 °C, ^1^H NMR (400 MHz, Acetone-*d*_6_, TMS): δ 9.24 (br, 2H,-OH), 7.60 (d, *J* = 15.8 Hz, 2H, Vinyl-H), 7.55 (m, 4H, Aryl-H), 6.90 (m, 4H, Aryl-H), 6.66 (d, *J* = 15.8 Hz, 2H, Vinyl-H), 5.99 (s, 1H, Methine-H); ^13^C NMR (100 MHz, Acetone-*d*_6_, TMS): δ 184.57 (C=O), 160.77 (C-OH), 141.19 (C_vinyl_-H), 131.03 (C_aryl_-H), 127.58 (C_aryl_), 121.98 (C_vinyl_-H), 116.86 (C_aryl_-H), 101.78 (C_methine_-H). IR (KBr) 3232 v(OH), 1622 v(C=O), 1599 v( C=O), 1513 v(C=O, C=C), 1444 v(OH), 1276 v(C-O_enol_), 1140 v(C-O)cm^−1^, EI-MS: *m/z* = 308, *m/z* calc. = 308.

Furan-curcuminoid (7) 1,7-di(furan-2-yl)-5-hydroxyhepta-1,4,6-trien-3-one, brown powder, purified by recrystallization using EtOAc and hexane, purity by HPLC 99.33%, melting point 130 °C, ^1^H NMR (400 MHz, DMSO-*d*_6_, TMS): δ 16.06 (br, 1H, Enol-H), 7.87 (d, *J* = 1.70, 2H, Aryl-H), 7.45 (d, *J* = 15.71, 2H, Vinyl-H), 6.96 (d, *J* = 3.37, 2H, Aryl-H), 6.66 (dd, *J* = 3.43; 1.68, 2H, Aryl-H), 6.57 (d, *J* = 15.71, 2H, Vinyl-H), 6.19 (s, 1H, Methine-H); ^13^C NMR (100 MHz, DMSO-*d*_6_, TMS): δ 182.48 (C=O), 151.02 (C_aryl_), 146.12 (C_aryl_-H), 126.94 (C_vinyl_-H), 121.18 (C_vinyl_-H), 116.15 (C_aryl_-H), 113.03 (C_aryl_-H), 101.98 (C_methine_-H). IR (KBr) 3124 v(C=C_ring_), 1628 v(C=O), 1563 v(C=O, C=C), 1468 v(C=C_ring_), 1262 v(C-O_enol_), 1139 v(C-O), 962 v(C-C=C) cm^−1^, EI-MS: *m/z* = 256, *m/z* calc. = 256.

Thiophene-curcuminoid (8) 1,7-di(thiophen-2-yl)-5-hydroxyhepta-1,4,6-trien-3-one, yellow powder, purified by recrystallization using acetone and hexane, purity by HPLC 98.68%, melting point 184 °C, ^1^H NMR (400 MHz, DMSO-*d*_6_, TMS): δ 16.07 (br, 1H, Enol-H), 7.81 (d, *J* = 15.71, 1H, Vinyl-H), 7.75 (d, *J* = 5.04, 2H, Aryl-H), 7.54 (d, *J* = 3.31, 2H, Aryl-H), 7.17 (dd, *J* = 5.04; 3.60, 2H, Aryl-H), 6.56 (d, *J* = 15.64, 2H, Vinyl-H), 6.19 (s, 1H, Methine-H); ^13^C NMR (100 MHz, DMSO-*d*_6_, TMS): δ 182.52 (C=O), 139.83 (C_aryl_), 133.23 (C_vinyl_-H), 132.06 (C_aryl_-H), 130.06 (C_aryl_-H), 128.76 (C_aryl_-H), 122.70 (C_vinyl_-H), 101.52 (C_methine_-H). IR (KBr) 3102 v(C=C_ring_), 1619 v(C=O), 1565 (C=O, C=C), 1505 v(C-O), 1418 v(C-OH), 964 (C-C=C) cm^−1^, EI-MS: *m/z* = 288, *m/z* calc. = 288.

## 5. Conclusions

Using simple high-yield steps, we contribute with a laboratory-scale strategy to obtain curcumin and symmetric curcuminoids. As a result, it can provide significant quantities of these compounds for physicochemical, analytical, and biological assay studies. This synthetic route is appropriate for using different aldehydes to obtain the corresponding symmetric curcuminoids. Due to the accessibility of this simple three-step synthetic approach, the method offers excellent potential for making available curcumin and symmetric curcuminoids on a large scale. The benefits of the present synthesis widen the perspectives for expanding the scientific studies concerning the fascinating molecular structures of curcuminoids and their widely recognized biological effects. 

## 6. Patents

An application for a patent is underway in the country of the authors.

## Data Availability

Not applicable.

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
