# Peer review of "High Yield Synthesis of Curcumin and Symmetric Curcuminoids: A “Click” and “Unclick” Chemistry Approach"

_molecules, 2022, doi:10.3390/molecules28010289_

Round 1

Reviewer 1 Report

The MS entitled “Click and Unclick Chemistry in the synthesis of Curcumin and symmetric curcuminoids” was thoroughly reviewed. The article is acceptable however, the following suggestions/corrections must be addressed by the authors before further decision.

1.      Title of the study is somewhat different from the core objective of the study; it should be modified according to the main results of this work. Moreover, the author should briefly explain in a para the significances of Click and Unclick chemistry in introduction section.

2.      The abstract should be started with the initial sentences that reflect the medicinal properties of curcumin and curcuminoids from the literature.

3.      The abstract should be amended with some numerical values from the key results.

4.      Key words; the author has mention “anti-cancer” while there was no anti-cancer activity found in this manuscript neither in methods nor in results, in my opinion the author should remove it from section Key words.

5.      If the authors have performed any anti-cancer activity of these synthesized compounds, then mention it in methods and results section of the manuscript.

6.      In introduction section: line 30 to 34, in description of curcumin, it is suggested to incorporate and elaborate further by supporting more relevant and latest updated research articles; for guidance;

https://doi.org/10.3390/biomedicines10102597,

https://doi.org/10.3390/biomedicines10102385. 

https://doi.org/10.3390/antibiotics11020135.

 https://doi.org/10.3390/molecules27082468.

https://doi.org/10.1177/1934578X0600100613.

DOI: 10.1016/j.jtcme.2016.05.005.

7.      Line 61, 62 seems incomplete, revise and rephrase to be readable.

8.      The author should provide the rational for designing curcumin and curcuminoids in the last para of introduction.

9.      It would be better to shift scheme 3, 4, and table 2, 3 from discussion section and place in results section in appropriately.

10.   Methods in general lack references and should be supported by appropriate references.

11.   The discussion section should be initiated with para showing significances of the study.

12.   Most of the synthesis procedures in methods are placed without mentioning the previous work. It would be better to provide supportive references.

13.   In methods: Most of the methods lack references. Here each section should be supported by suitable appropriate reference to make it more authentic.

14.   The article should be revised thoroughly for grammatical mistakes.

Author Response

The MS entitled ““Click and Unclick Chemistry in the synthesis of Curcumin and symmetric curcuminoids” was thoroughly reviewed. The article is acceptable however, the following suggestions/corrections must be addressed by the authors before further decision.

Dear colleague Referee 1:

Below you will find point by point our

  1. Title of the study is somewhat different from the core objective of the study; it should be modified according to the main results of this work. Moreover, the author should briefly explain in a para the significances of Click and Unclick chemistry in introduction section.

The title has been modified considering your suggestion, although we suggest that we preserve the terms “click” and “unclick” since it strictly fulfills the spirit with which the term “click” was originally coined long ago by Prof. Sharpless. The antonym term “unclick” also adheres to the same spirit of reaction in the removal of the regioselective protector/activator group BF2. Both terms give merit to the ease and high efficiency of these single step reactions involved. In addition, the use of the less common term “uncllck” is intended to add a bonus of attraction to readers while preserving grammar rigor. Last, we avoid referring to only two heterocyclic symmetric heterocyclic derivatives since our undergoing experimentation is revealing a wide applicability that will be seen in the short future. We agree completely with you that the title plays a very important role. Therefore, we trust your final decision regarding this point.

“High Yield Synthesis of Curcumin and Symmetric Curcuminoids: A “Click” and “Unclick” Chemistry Approach”.

  1. The abstract should be started with the initial sentences that reflect the medicinal properties of curcumin and part of curcuminoids from the literature.

We appreciate the observation; therefore, the abstract has been modified with the suggested request.

  1. The abstract should be amended with some numerical values from the key results.

To improve the abstract, numerical data were added where applicable.

  1. Key words; the author has mention “anti-cancer” while there was no anti-cancer activity found in this manuscript neither in methods nor in results, in my opinion the author should remove it from section Key words.

The keyword “anti-cancer” was removed according to your suggestion.

  1. If the authors have performed any anti-cancer activity of these synthesized compounds, then mention it in methods and results section of the manuscript.

In fact, in this work, an anti-cancer study was not carried out; and we emphasize the possible biological activity of curcuminoids in accordance with numerous previous reports.

  1. In introduction section: line 30 to 34, in description of curcumin, it is suggested to incorporate and elaborate further by supporting more relevant and latest updated research articles; for guidance;

https://doi.org/10.3390/biomedicines10102597,

https://doi.org/10.3390/biomedicines10102385. 

https://doi.org/10.3390/antibiotics11020135.

 https://doi.org/10.3390/molecules27082468.

https://doi.org/10.1177/1934578X0600100613.

DOI: 10.1016/j.jtcme.2016.05.005.

We add the references suggested that reinforce the biological significance of curcumin and its derivatives. The references added correspond now to references 25, 31, 36, 40, 41, and 45. These references are useful and relevant indeed.

  1. Line 61, 62 seems incomplete, revise and rephrase to be readable.

Lines 61 and 62 were rephrased in order to improve grammar and English scientific usage.

  1. The author should provide the rational for designing curcumin and curcuminoids in the last para of introduction.

Based on your comment, we added a paragraph on the design of the present curcuminoid compounds.

  1. It would be better to shift scheme 3, 4, and table 2, 3 from discussion section and place in results section in appropriately.

Schemes 3 and 4 and tables 2 and 3 were placed in the results section. We appreciate the suggestions about moving schemes and tables of this work.

  1. Methods in general lack references and should be supported by appropriate references.

We appreciate the observation made. We add the appropriate references of each method process

  1. The discussion section should be initiated with para showing significances of the study.

In the discussion of results, we added a paragraph dealing with this aspect of our work.

  1. Most of the synthesis procedures in methods are placed without mentioning the previous work. It would be better to provide supportive references.

We added references of previous work on which the present investigation is based.

  1. In methods: Most of the methods lack references. Here each section should be supported by suitable appropriate reference to make it more authentic.

We have added now appropriate references for each procedure of the methods.

  1. The article should be revised thoroughly for grammatical mistakes.

We appreciate the suggestion. We review all the text to eliminate editorial errors. We sincerely appreciate your time dedicated to improve our manuscript.

Reviewer 2 Report

I suggest to change the title:
Synthesis of Curcumin and 2 symmetric curcuminoids

The description of the spectrum on Figure 1,2,3 is incomprehensible. Please to connect all signals with respective protons. Next, the coupling constants should be illustrated and defined.

Table 1 and 2:
"overnight" - is is not precise. please define the time in hours.

Table 2:
"partial reaction" - this is unclear.

The mechanism presented on Scheme 2 is supported by any way?

Figure 4 should be removed.

Paragraph 4:
Conditions and technical details for MS experiments should be added.

Paragraph 4:
Type of column for HPLC should be precised.

Descriptions of synthetised compounds:
IR signals should be defined.
NMR signals should be connected with respective atoms.

Author Response

I suggest to change the title:
Synthesis of Curcumin and 2 symmetric curcuminoids

Dear referee 2:

Below you will find all modifications made to our manuscript following as close as possible your useful recommendations.

The title has been modified considering your suggestion, but it was also modified keeping the spirit of the article with the aim of making it more attractive and objective to readers.

“High Yield Synthesis of Curcumin and Symmetric Curcuminoids: A “Click” and “Unclick” Chemistry Approach”. We respectfully share with you and referee 1, the paragraph below concerning the title of our manuscript. Since we recognize the importance of a good objective, yet attractive title, please note that we will definitively follow your advice concerning this point.

The title has been modified considering your suggestion, although we suggest that we preserve the terms “click” and “unclick” since it strictly fulfills the spirit with which the term “click” was originally coined long ago by Prof. Sharpless. The antonym term “unclick” also adheres to the same spirit of reaction in the removal of the regioselective protector/activator group BF2. Both terms give merit to the ease and high efficiency of these single step reactions involved. In addition, the use of the less common term “uncllck” is intended to add a bonus of attraction to readers while preserving grammar rigor. Last, we avoid referring to only two heterocyclic symmetric heterocyclic derivatives since our undergoing experimentation is revealing a wide applicability that will be seen in the short future. We agree completely with you that the title plays a very important role. Therefore, we trust your final decision regarding this point. Proposed new title: “High Yield Synthesis of Curcumin and Symmetric Curcuminoids: A “Click” and “Unclick” Chemistry Approach”.

The description of the spectrum on Figure 1,2,3 is incomprehensible. Please to connect all signals with respective protons. Next, the coupling constants should be illustrated and defined.

Dear reviewer,  Figures 1, 2 and 3 have been correctly modified according as requested; the coupling constants included are as well as the assignments of each molecule (see Manuscript)

Table 1 and 2:
"overnight" - is is not precise. please define the time in hours.

It has been decided to change the times of table 1 and 2, from hours instead of overnight

Table 2:
"partial reaction" - this is unclear.

the appropriate precision has been taken, instead, the reaction has been refered to in terms of percentage.

The mechanism presented on Scheme 2 is supported by any way?

the mechanism was supported by appropriate references and are indicated in the manuscript

Figure 4 should be removed.

the mass spectrum in Figure 4 has been deleted from the manuscript and appropriately moved to the supplementary material as suggested.

Paragraph 4:
Conditions and technical details for MS experiments should be added.

The experimental conditions for obtaining the mass spectra is now in appropriate usage and is described in the manuscript (see materials and methods section).

Paragraph 4:
Type of column for HPLC should be precised.

A 100 x 2.0 mm 3 μm Eclipse Plus C18(2) column was used, and it has already been indicated in the manuscript (see materials and methods section).

Descriptions of synthetised compounds:
IR signals should be defined.
NMR signals should be connected with respective atoms.

the IR and NMR signals have been assigned as requested and based on the journal´s format; the connectivity of the atoms has been performed for each NMR signal (see general methodology and supplementary material section).

Round 2

Reviewer 1 Report

The manuscript is improved a lot. I have no issue with the title. Check for final spellings and grammar before publication.